# Animals as Architects: Building the Future of Technology-Supported Rehabilitation with Biomimetic Principles

**DOI:** 10.3390/biomimetics9120723

**Published:** 2024-11-22

**Authors:** Bruno Bonnechère

**Affiliations:** 1REVAL Rehabilitation Research Center, Faculty of Rehabilitation Sciences, Hasselt University, 3590 Diepenbeek, Belgium; bruno.bonnechere@uhasselt.be; 2Technology-Supported and Data-Driven Rehabilitation, Data Sciences Institute, Hasselt University, 3590 Diepenbeek, Belgium; 3Department of PXL—Healthcare, PXL University of Applied Sciences and Arts, 3500 Hasselt, Belgium

**Keywords:** rehabilitation technology, animal models, prosthetics, neural plasticity, assistive technology, regenerative medicine

## Abstract

Rehabilitation science has evolved significantly with the integration of technology-supported interventions, offering objective assessments, personalized programs, and real-time feedback for patients. Despite these advances, challenges remain in fully addressing the complexities of human recovery through the rehabilitation process. Over the last few years, there has been a growing interest in the application of biomimetics to inspire technological innovation. This review explores the application of biomimetic principles in rehabilitation technologies, focusing on the use of animal models to help the design of assistive devices such as robotic exoskeletons, prosthetics, and wearable sensors. Animal locomotion studies have, for example, inspired energy-efficient exoskeletons that mimic natural gait, while insights from neural plasticity research in species like zebrafish and axolotls are advancing regenerative medicine and rehabilitation techniques. Sensory systems in animals, such as the lateral line in fish, have also led to the development of wearable sensors that provide real-time feedback for motor learning. By integrating biomimetic approaches, rehabilitation technologies can better adapt to patient needs, ultimately improving functional outcomes. As the field advances, challenges related to translating animal research to human applications, ethical considerations, and technical barriers must be addressed to unlock the full potential of biomimetic rehabilitation.

## 1. Introduction

Rehabilitation is a fundamental component of healthcare that focuses on helping individuals recover, improve, or maintain functional abilities following injury, surgery, or the onset of chronic conditions [1]. Traditionally, rehabilitation has relied on therapist-guided exercises and manual techniques to restore patient function [2]. However, the field has undergone a significant transformation in recent years, with the integration of technology-supported interventions that provide more objective assessments, personalized interventions, and real-time feedback for patients [3]. Technologies such as virtual reality (VR) systems, robotic exoskeletons, wearable sensors, and mobile health applications have become integral components of modern rehabilitation, enabling patients to engage in more effective and adaptive therapeutic programs [4].

The rapid advancement of these technologies has opened new possibilities for enhancing recovery, especially for individuals with neurological and musculoskeletal impairments. Yet, despite these advances, there is an emerging consensus that technology alone cannot fully address the intricacies of human movement and the complexities of the recovery process [5]. This has led to a growing interest in biomimetics—the study of nature’s models, systems, and elements to inspire new technologies [6]. Biomimetic approaches in rehabilitation focus on understanding and replicating the highly efficient, adaptive mechanisms found in animals to develop therapeutic solutions that can better support human recovery.

Animal models play a crucial role in this biomimetic approach, offering unique insights into the mechanisms of movement, sensory processing, and recovery from injury. Animals have long played a pivotal role in biomechanics and motion analysis, a practice that dates back to the 19th century. A notable figure in this field, Eadweard Muybridge, revolutionized motion capture through his pioneering photographic techniques [7]. Muybridge’s work began when he was tasked with proving whether all four of a horse’s hooves left the ground during a gallop. In 1878, he successfully captured this in a series of photographs using a system of cameras triggered in rapid succession (Figure 1). His study, known as The Horse in Motion, became one of the earliest examples of biomechanical motion analysis. Muybridge’s methods laid the foundation for modern techniques in motion capture, which are now integral in studying both human and animal locomotion [8].

A few years later, in 1885, Muybridge and Dercum, working at the University of Pennsylvania, conducted the first motion picture study of neurological patients using early photographic technology previously developed [9]. The technological advancements they initiated, including the use of high-speed photography, enabled researchers to dissect the biomechanics of movement in ways that were previously impossible (Figure 2), creating a pathway for the use of animals in gait and motion studies [10]. Studying the gait of animals like mice has provided invaluable data on how central nervous system (CNS) dysfunctions impact movement patterns. In a translational study comparing gait signatures between mice and humans, researchers have developed a method to account for changes in walking speed and velocity, enabling a more accurate comparison of gait abnormalities across species [11]. This approach is significant as it helps bridge the gap between quadrupedal and bipedal locomotion, overcoming one of the major barriers in translating findings from animal models to human applications. By understanding how gait parameters change in response to neurological conditions like Parkinson’s disease in both human and animal models, researchers can better identify the underlying CNS circuit dysfunctions and develop targeted treatment strategies.

The value of animal models extends beyond mere observation of movement. Studies using mice with induced conditions such as experimental parkinsonism have revealed how changes in gait quality and stride length can indicate shifts in the neurological control of movement [11]. These findings are not just theoretical; they have practical implications for designing rehabilitation technologies that can adjust to the specific needs of patients. For example, robotic gait trainers and exoskeletons can be programmed to mimic the natural gait patterns observed in animals, adapting to changes in stride length, stance duration, and cadence to provide a more effective rehabilitation experience. The insights from these animal studies ensure that rehabilitation devices are not simply rigid tools but adaptive systems that respond to the complexities of human movement in a dynamic way.

Another area where animal models have significantly contributed is in understanding the relationship between body size, posture, speed, and movement efficiency. Predictive musculoskeletal simulations have been used to study a wide range of mammalian sizes, from small animals like mice to larger species like elephants, simulating how these different sizes impact maximum speed and energy efficiency during movement [12]. The simulations reveal a curious pattern where intermediate-sized animals achieve the fastest speeds, with a notable decline in speed among both the smallest and largest species. This hyperbolic relationship highlights the balance between muscle force production, ground reaction forces, and the mechanical constraints of locomotion, which are critical for optimizing prosthetic and orthotic designs in humans. By applying these principles, engineers can create more efficient and effective mobility aids that take into account the biomechanics of movement, allowing users to conserve energy and achieve more natural gait patterns.

The findings from these studies have profound implications for rehabilitation, especially in the context of designing assistive technologies that are both energy-efficient and effective at restoring natural movement. For instance, robotic exoskeletons inspired by the spring-like mechanisms found in animals such as kangaroos can store and release energy during movement [13], reducing the metabolic cost of walking for users and enhancing their endurance [14]. Similarly, wearable sensors that emulate the proprioceptive feedback systems found in animals, like the lateral line system in fish, provide real-time data on body movements, enabling precise adjustments during rehabilitation exercises. This approach ensures that patients receive immediate feedback, which is critical for motor learning and improving functional outcomes during recovery [15].

Moreover, animal models have also been instrumental in advancing the field of neural regeneration and plasticity, which are essential for recovery from neurological injuries such as strokes or spinal cord injuries [16]. Species like the zebrafish and axolotl are known for their remarkable ability to regenerate spinal cord neurons and even entire limbs after injury [17]. Research into the molecular pathways that enable these regenerative processes has provided valuable insights that could be applied to human therapy, potentially leading to new treatments that promote neural plasticity and facilitate recovery in patients with CNS injuries.

In summary, the integration of biomimetic principles into technology-supported rehabilitation represents a promising path forward for enhancing therapeutic strategies. By leveraging insights gained from animal models, researchers can develop innovative rehabilitation technologies that are better suited to the complex needs of human patients. The study of animal locomotion, neural adaptation, and energy-efficient movement patterns offers a roadmap for designing devices that not only mimic natural processes but also improve functional outcomes for patients. As the field continues to evolve, it is increasingly clear that a deeper understanding of nature’s solutions will be key to unlocking the full potential of rehabilitation technology, offering new hope and possibilities for individuals on their journey to recovery.

## 2. Animal Models in (Rehabilitation) Research

Animal models have long served as a cornerstone in biomedical research, providing unique insights into the physiological and behavioral adaptations that have evolved over millions of years [18], some examples (non-exhaustive list) are presented in Table 1.

These models are particularly valuable for understanding the mechanisms of movement, sensory processing, and recovery from injury, making them indispensable in the field of rehabilitation research. By studying the ways animals adapt to their environments, including their locomotion patterns, researchers have uncovered principles that can be applied to human rehabilitation—especially in developing technologies that emulate or enhance natural processes. This section explores how animal models have been leveraged to advance rehabilitation science, focusing on key areas such as locomotion and prosthetics, neural plasticity and recovery, and sensory systems and assistive technologies.

### 2.1. Locomotion and Prosthetics

Animal locomotion has not only provided insight for motion analysis [7], but also provided inspiration for developing prosthetic and orthotic devices aimed at restoring movement in humans. The efficiency and adaptability observed in animal movement offer valuable models for creating prosthetics that more closely mimic natural human gait [29]. For example, studies on the gait of quadrupeds, such as horses and dogs, have informed the design of lower limb prosthetics that minimize energy expenditure during walking, allowing users to move with greater ease and endurance [30].

A particularly important area of research has been the analysis of gait changes in animals, such as mice, after inducing neurological conditions like parkinsonism. These studies help to understand how CNS dysfunctions alter gait parameters such as stride length, cadence, and stance duration [31]. For instance, research has shown that parkinsonism in mouse models leads to significant shifts in gait patterns, which can be studied through velocity-dependent regression models [11]. These findings are crucial for human applications because they allow researchers to differentiate between changes in gait speed and changes in gait quality, helping to refine the design of therapeutic interventions that target specific gait abnormalities in humans, such as robotic gait trainers and exoskeletons [32].

While functional biomimetics emphasizes solving specific problems by mimicking nature’s solutions, esthetic biomimetics often copies natural movements for appearance rather than performance. This distinction is particularly relevant in prosthetic design. Although mimicking smooth human movements (such as through the minimum jerk model) is popular in synthetic systems, this approach may not optimize the control objectives like speed and accuracy in prosthetics [33]. True biomimetic systems must account for the complex neural and muscular architecture that governs human movement. Therefore, future innovations in prosthetics may benefit from integrating neuro-morphic systems that adapt to natural plasticity, enabling prosthetic limbs to achieve human-like movement over time [33].

Robotic exoskeletons, inspired by the biomechanics of various animals, have become a key application of this research. These devices are designed to assist individuals with mobility impairments by augmenting their strength and endurance during walking [34]. For example, exoskeletons that emulate the energy-storing capabilities of a kangaroo’s hind legs provide a more natural gait experience for users, facilitating longer and more effective rehabilitation sessions [13,35]. By incorporating the principles of animal locomotion, these technologies can adapt to the user’s movement needs, allowing for a more seamless transition between different walking speeds and terrains, similar to the adaptability seen in animal gait patterns [36].

### 2.2. Neural Plasticity and Recovery

Neural plasticity, or the ability of the brain and nervous system to adapt and reorganize itself, is a fundamental factor in recovery from neurological injuries, such as stroke and spinal cord damage [37]. Animals exhibit extraordinary neural plasticity, making them valuable models for studying processes that could enhance similar capabilities in humans. For example, zebrafish have the ability to regenerate spinal cord neurons after injury, a capability that has made them a model organism for studying spinal cord regeneration and repair [38]. Research into the molecular mechanisms underlying this regenerative process has provided insights that could inform new treatments aimed at promoting neural regeneration and plasticity in humans, potentially leading to breakthroughs in therapies for spinal cord injuries and neurodegenerative diseases [39].

Another notable example is the axolotl, a salamander species known for its ability to regenerate not only limbs but also spinal cord tissue and even portions of its brain [40]. This remarkable capacity is linked to the axolotl’s retention of embryonic characteristics throughout its life, a phenomenon known as neoteny. Understanding the molecular pathways that enable the axolotl’s regenerative abilities has inspired research into methods for enhancing neural plasticity and tissue regeneration in humans [41]. These studies provide a roadmap for developing therapies that could facilitate recovery from conditions such as traumatic brain injuries or degenerative neurological disorders by harnessing similar regenerative processes [42].

The findings from these animal studies have practical implications for rehabilitation technologies. For example, robotic interfaces and neuroprosthetics designed to support recovery from spinal cord injuries can be adapted based on insights gained from the regenerative processes observed in animals. Such devices can be programmed to encourage neural pathways to reorganize and adapt, aiding in the restoration of motor function in patients undergoing rehabilitation [43].

### 2.3. Sensory Systems and Assistive Technologies

The sensory systems of animals are highly specialized, enabling them to interact with their environments in ways that often surpass human capabilities [44]. These systems have served as models for developing assistive technologies that enhance or compensate for sensory impairments in humans [45]. A prime example is the echolocation ability of bats, which has inspired the development of devices for visually impaired individuals [46]. These devices use ultrasonic waves to map the surrounding environment, providing auditory feedback that allows users to navigate spaces with greater ease and confidence [47].

Similarly, the lateral line system found in fish, which detects subtle changes in water currents and vibrations [48], has influenced the design of wearable sensors for rehabilitation [49]. These wearable systems can provide real-time feedback on body posture and movement, similar to how fish sense their surroundings. By emulating the proprioceptive feedback mechanisms of the lateral line system, these devices allow for precise corrections during rehabilitation exercises, enhancing motor learning and improving balance and coordination in patients recovering from strokes or traumatic brain injuries. Clinical studies have shown that patients who use such biomimetic devices often demonstrate better functional outcomes compared to those undergoing traditional rehabilitation methods [50].

Moreover, the electroreception abilities of certain fish, such as sharks and rays, have inspired sensory augmentation devices that aim to enhance the input available to individuals with sensory processing disorders [51]. These technologies can provide alternative means of environmental interaction for people with severe sensory impairments, offering new possibilities for improving their quality of life.

### 2.4. Brain Machine Interface

Brain-machine interfaces (BMI) have emerged as a promising technology to restore communication pathways between the nervous system and external devices, offering promising advancements in physical medicine and rehabilitation aimed at recovering lost sensory and motor functions [52]. However, despite significant progress in basic research, the majority of the systems developed in laboratory settings have not yet successfully transitioned into practical, home-based applications [53]. One of the primary challenges hindering this shift is the absence of naturalistic feedback mechanisms that inform users of the consequences of their BMI-controlled actions.

To address this limitation, recent BMI advancements have been guided by the principle of biomimicry—replicating natural neural processes to enhance the functionality of these systems. The focus on reproducing biological feedback mechanisms, particularly somatosensory feedback, has become crucial for improving the user experience and effectiveness of BMIs. Somatosensory feedback, which provides information about body position, movement, and interaction with the environment, is key in enabling the more natural and intuitive control of BMIs. By integrating such feedback, BMIs can offer users a more realistic and precise interaction with their external environment [54].

Advancements in both invasive and non-invasive BMI systems have shown varying degrees of biomimicry. Invasive BMIs, which involve direct implantation of electrodes into the brain, often provide more precise control and feedback but come with higher risks and complexity. Non-invasive BMIs, which rely on external sensors, are safer and more accessible but may offer less precise feedback and control. The challenge lies in striking a balance between the level of biomimicry achieved and the practicality of the BMI system for everyday use [54].

A notable approach to enhancing BMIs involves the integration of sensory feedback that mimics natural touch or proprioception [55]. In animal studies, for example, researchers have used primates to explore the effectiveness of artificial sensory feedback in BMI systems. In these studies, monkeys have been able to perform complex tasks with a BMI by receiving tactile or proprioceptive feedback through artificial means, allowing them to adjust their movements more accurately. These findings underscore the importance of somatosensory feedback in making BMIs more intuitive and effective, offering insights into how similar systems could be applied to human rehabilitation [54].

## 3. Case Studies in Biomimetic Rehabilitation Technologies

### 3.1. Selection Criteria and Unique Contributions of Animal Models in Biomimetic Rehabilitation Technologies

The selection of animal models in biomimetic rehabilitation research is guided by specific criteria that maximize the relevance of findings for human applications. Key considerations include anatomical and physiological similarities to humans, unique regenerative capabilities, neural plasticity, and movement patterns that align closely with human biomechanics. For instance, species like zebrafish and axolotls are frequently chosen for their remarkable regenerative abilities, offering insights into neural and spinal cord recovery mechanisms. These models contribute significantly to the development of technologies aimed at promoting neural regeneration and functional recovery following spinal injuries in humans. The integration of insights from animal models into the design of rehabilitation technologies is not merely theoretical but has led to tangible advancements in patient care; a few examples are presented in Table 2 and Figure 3.

Quadrupedal animals such as dogs and horses provide valuable data on gait dynamics, which are integral to the design of prosthetic limbs and robotic exoskeletons. Their gait patterns and movement efficiency offer insights into energy conservation strategies that can be replicated in rehabilitation devices, enhancing their adaptability and energy efficiency for users [56]. Additionally, rodent models, particularly those of mice and rats, are extensively used to study neurological impairments and motor function recovery. These models facilitate the study of central nervous system dysfunctions and gait abnormalities, which directly inform the design of neurorehabilitation technologies for conditions such as stroke or Parkinson’s disease [57,58].

The proprioceptive and sensory systems of certain animals have also contributed unique insights. For example, the lateral line system in fish has inspired the development of wearable sensors that provide proprioceptive feedback, enhancing motor learning and correction during rehabilitation [59]. This feedback-driven approach is critical in rehabilitation technologies that aim to restore balance and coordination in patients recovering from neurological injuries.

**Table 2 biomimetics-09-00723-t002:** Examples of animal studies in rehabilitation research.

Animal Model	Research Area	Purpose/Insight	Examples/Applications
Rats	Spinal Cord Injury [60]	Used to study nerve regeneration and motor recovery through rehabilitation techniques such as treadmill training and electrical stimulation.	Development of therapies to enhance motor function recovery after spinal cord injuries.
Mice	Stroke Recovery [61]	Models for studying brain plasticity and motor recovery, allowing testing of rehabilitation strategies like constraint-induced movement therapy.	Insights into neuroplasticity and development of post-stroke rehabilitation protocols.
Cats	Locomotor Rehabilitation [62]	Employed in studies of gait and locomotor function, particularly useful in understanding how spinal circuits contribute to movement recovery.	Development of robotic-assisted gait training devices for humans with spinal injuries.
Pigs	Muscle Regeneration [63]	Used to study the effects of exercise and physical therapy on muscle recovery after injury due to their muscle structure similarity to humans.	Insights into improving rehabilitation for muscle injuries, such as post-surgical recovery.
Non-Human Primates (e.g., macaques)	Cognitive Rehabilitation [64]	Models for testing rehabilitation approaches for cognitive and motor function recovery after brain injuries, closely mimicking human brain function.	Development of cognitive therapies and advanced rehabilitation technologies.
Rabbits	Tendon Healing [65]	Studied for their response to various physical therapy modalities like ultrasound and controlled movements to enhance tendon repair.	Development of rehabilitation protocols for tendinitis and post-surgical tendon recovery.
Dogs	Orthopedic Rehabilitation [66]	Used for modeling rehabilitation after joint surgeries, including physical therapy regimens to restore range of motion.	Insights into physical therapy techniques for post-operative care in joint replacement.
Zebrafish	Regenerative Medicine [67]	Allows study of neural regeneration and recovery due to their ability to regenerate central nervous system tissues.	Development of approaches for nerve regeneration and spinal injury recovery in humans.
Sheep	Joint Rehabilitation [68]	Used to study cartilage repair and recovery after joint injuries, focusing on rehabilitation techniques that promote healing.	Insights into physical therapy methods for joint recovery and treatment of osteoarthritis.
Rats	Peripheral Nerve Injury [69]	Models for studying the recovery of sensory and motor functions after nerve damage using rehabilitation strategies like electrical stimulation.	Insights into therapies that aid nerve recovery and improve sensory function after injuries.

By carefully selecting animal models based on these specific contributions, researchers can ensure that biomimetic principles are effectively translated into functional rehabilitation technologies that meet the complex needs of human patients.

By studying how animals maintain stability and adapt to varying speeds and terrains [12], researchers can design rehabilitation devices that are not only effective but also adaptable to real-world environments. This adaptability is critical for ensuring that patients can translate the skills they acquire in clinical settings to everyday life, leading to more sustainable recovery outcomes. As rehabilitation research continues to evolve, the integration of biomimetic principles promises to create more effective and personalized care strategies that bring the benefits of nature’s innovations to those in need of recovery.

This section presents several case studies that highlight how insights from animal models have been successfully translated into practical rehabilitation technologies, illustrating the potential of biomimetics to improve rehabilitation.

### 3.2. Practical Examples

#### 3.2.1. Robotic Exoskeletons

Robotic exoskeletons represent one of the most promising biomimetic technologies in the field of rehabilitation, particularly for patients recovering from spinal cord injuries, strokes, or other conditions that impair mobility [70]. These devices are designed to augment the user’s physical capabilities by providing additional support during movement, enabling individuals to regain lost function through repetitive, assisted gait training. Many exoskeletons are designed to mimic the energy-conserving mechanics of kangaroos and other animals by incorporating spring-like mechanisms that store and release energy. These systems can be described using a simple spring-mass model, where the elastic potential energy (*U*) in the spring is given by Equation (1).
(1)$$U=\frac{1}{2}kx^{2}$$
where *k* is the spring constant and *x* is the displacement. This stored energy can be released to aid movement, thus reducing the metabolic energy required by the user.

A crucial element in the design of these exoskeletons is the incorporation of principles derived from animal locomotion, which ensures that the devices can replicate natural movement patterns and optimize energy efficiency [71].

Exoskeletons often use dynamic models that incorporate Lagrangian mechanics to optimize energy storage and release. For instance, the Lagrangian function *L* for a spring-mass system in an exoskeleton leg can be defined by Equation (2), as follows:(2)$$L=T-U=\frac{1}{2}mx^{2}-\frac{1}{2}kx^{2}$$
where *T* is the kinetic energy, *U* is the potential energy, *m* is the mass, *x*^2^ is the velocity, and *k* is the spring constant. By differentiating the Lagrangian function with respect to x and *x*^2^, equations of motion can be derived to optimize minimal energy expenditure and efficient gait assistance.

For example, the mechanics of animal locomotion have inspired the development of exoskeletons that mimic the energy-storing capabilities found in species like kangaroos and horses [72]. These exoskeletons utilize elastic components that store and release energy during walking, similar to the spring-like mechanisms of the hind legs. This design allows users to move more naturally, reducing the metabolic cost of walking and increasing their endurance during rehabilitation sessions [73]. The ability to replicate these energy-efficient movement strategies is particularly valuable for patients who may have limited physical strength, as it enables them to engage in more intensive and prolonged training sessions, which are critical for successful rehabilitation outcomes.

Proportional-derivative control is often used to maintain stability and responsiveness. The control force *F* applied by the exoskeleton can be defined by Equation (3), as follows:(3)$$F=K_{p}\left(x_{d}-x\right)+K_{d}({\dot{x}}_{d}-\dot{x})$$
where $K_{p}$ and $K_{p}$ are the proportional and derivative gains, $x_{d}$ is the desired position, and ${\dot{x}}_{d}$ is the desired velocity. These parameters are tuned based on the biomechanics of natural gait, ensuring that the exoskeleton’s movement closely matches the user’s; an example is presented in Figure 4.

In rehabilitation, an anthropomorphic exoskeleton with a variable instantaneous center of rotation has been proposed to accommodate the individual differences in knee joint mechanics. The exoskeleton design utilizes a two-degrees-of-freedom, five-bar mechanism optimized to reduce errors in knee biomechanics, gait angle, and ankle trajectory, achieving near-human movement patterns. Experimental tests confirmed minimal discrepancies, indicating improved comfort and reduced human–machine interaction forces during use [50]. Additionally, a novel central tendon-based bellows actuator inspired by the muscle structure of dragonflies has been integrated. This actuator significantly enhances tensile strength compared to traditional pneumatic actuators, offering over ten times the pulling force with minimal added weight, presenting promising applications in exoskeleton robotics for rehabilitation and beyond [51]. Advanced exoskeletons use variable stiffness actuators, allowing the adjustment of the spring constant k in response to movement demands. This stiffness modulation is calculated to adapt dynamically, enhancing energy conservation by imitating how biological muscles alter stiffness during different phases of gait.

In addition, exoskeletons that draw on the lightweight and adaptive movement strategies of birds and insects, thanks to modern 3D printing techniques, have been developed to support gait training in clinical settings [74].

Furthermore, advanced prosthetic limbs can use neuro-morphic control, which simulates human neural pathways, to enable real-time adaptability in response to the user’s movements. The control of these limbs can be optimized by minimizing the “jerk” (rate of change in acceleration), often modeled with the minimum jerk model Equation (4) as follows:(4)$$J=\int_{0}^{T}{\left(\frac{d^{3}x}{{dt}^{3}}\right)}^{2}dt$$
where *J* represents jerk, *x* is position, and *T* is the duration. Minimizing this integral allows for smoother, more human-like motion, which is especially useful in prosthetics intended for daily activities.

#### 3.2.2. Wearable Sensors and Feedback Systems

Wearable sensors have become increasingly popular in rehabilitation, offering a way to monitor patient progress and provide real-time feedback on body movements [75]. These devices have been significantly influenced by the sensory systems of animals, which have evolved to provide precise and rapid feedback about the environment. For example, the lateral line system in fish, which detects changes in water currents and vibrations, has inspired the design of wearable systems that emulate proprioceptive feedback mechanisms [49]. These sensors provide continuous data on body posture, joint angles, and movement patterns, allowing patients to make immediate adjustments during rehabilitation exercises [76].

One practical application of these biomimetic wearable sensors is in gait training for patients recovering from stroke or traumatic brain injury. These devices can provide feedback on gait symmetry, stride length, and weight distribution during walking exercises, helping patients correct abnormal movement patterns and improve their balance [77].

Wearable sensors combine accelerometers, gyroscopes, and magnetometers in an Inertial Measurement Unit (IMU) to track complex movements. By applying sensor fusion algorithms, such as complementary filtering or extended Kalman filtering, the system can estimate orientation and angular velocity accurately through Equation (5), as follows:(5)$$\theta_{k}=\left(1-\alpha \right)\left(\theta_{\left(k-1\right)}+\omega ∆t\right)+\alpha a_{k}$$
where $\theta_{k}$ is the estimated angle at time *k*, ω is the angular velocity, ∆t is the time step, α is a filter constant, and $a_{k}$ is the accelerometer measurement. These filters are essential for minimizing noise and improving the reliability of data in real-time applications.

The ability to receive immediate feedback enhances motor learning by reinforcing correct movements and minimizing compensatory behaviors that could hinder recovery. This approach has been shown to improve outcomes in patients, leading to better coordination and more natural walking patterns compared to traditional rehabilitation methods that lack real-time monitoring [78].

Machine learning algorithms are often applied to process sensor data and provide feedback to users. For example, a support vector machine classifier can differentiate between types of movements (e.g., correct vs. incorrect gait patterns) by training on labeled datasets. The classifier function f(x) is given by Equation (6).
(6)$$f\left(x\right)=sign\left(\sum_{i=1}^{N}\alpha_{i}y_{i}K\left(x_{i},x\right)+b\right)$$
where x is the input data, $\alpha_{i}$ and $y_{i}$ are model parameters, $K\left(x_{i},x\right)$ is the kernel function, and *b* is the bias term. This allows wearable devices to detect and correct posture and gait in real time.

In addition to improving the precision of movement training, wearable sensors inspired by animal sensory systems have been adapted for use in virtual reality (VR) environments [79]. By integrating wearable devices with VR systems, patients can engage in immersive rehabilitation exercises that simulate real-life challenges. For instance, the sensors can track the user’s movements and adjust the virtual environment accordingly, creating a more engaging and interactive rehabilitation experience [80]. This combination of wearable sensors and VR, or augmented reality (AR), is especially useful for patients who need to practice complex motor tasks, such as navigating through crowded spaces or climbing stairs, in a controlled and safe setting.

#### 3.2.3. Regenerative Medicine and Biomaterials

Biomimetics has also made significant contributions to the field of regenerative medicine, particularly in the development of biomaterials and scaffolds designed to promote tissue repair and regeneration [81]. The study of animals with remarkable regenerative capabilities, such as salamanders and zebrafish, has provided critical insights into the cellular and molecular mechanisms that enable these species to regenerate limbs and spinal cord tissue. These insights have been translated into the design of biomimetic scaffolds that mimic the natural extracellular matrix found in regenerative species, providing a supportive environment for cell growth and differentiation [82].

Biomimetic scaffolds use finite element analysis to simulate stress–strain characteristics and ensure durability. For example, the von Mises stress criterion, often applied in scaffold design, is given by Equation (7).
(7)$$\sigma_{v}=\sqrt{\frac{1}{2}\left({\left(\sigma_{1}-\sigma_{2}\right)}^{2}+{\left(\sigma_{2}-\sigma_{3}\right)}^{2}+{\left(\sigma_{3}-\sigma_{1}\right)}^{2}\right)}$$
where $\sigma_{1}$, $\sigma_{2}$, and $\sigma_{3}$ are principal stresses. This equation ensures that the scaffold can withstand physiological loads without failure.

One prominent example is the development of biomimetic scaffolds that are used to repair damaged nerves and promote spinal cord regeneration. These scaffolds are designed to replicate the structure and function of the extracellular matrix, providing a framework for new nerve cells to grow and establish connections [83]. In addition to their structural properties, these biomaterials often incorporate bioactive molecules that mimic the signaling cues found in regenerative animals. These molecules play a crucial role in guiding the regenerative process, promoting cell proliferation and migration to the injury site. Clinical trials have shown that these biomimetic scaffolds can significantly improve the recovery of nerve function in patients with spinal cord injuries, offering new hope for those with conditions that were once considered irreversible [84].

Moreover, the application of biomimetic principles in regenerative medicine extends to the development of bioengineered tissues for musculoskeletal repair. Inspired by the natural healing processes observed in animals, researchers have created scaffolds that support the regeneration of cartilage, skin, and other tissues [85]. These biomaterials are designed to degrade gradually as new tissue forms, ensuring that the regeneration process mimics the natural healing trajectory seen in species like zebrafish [86].

For example, biodegradable scaffolds are designed to degrade at a rate matching tissue regeneration. This degradation follows a first-order kinetic model, where the degradation rate *D* is given by Equation (8).
(8)$$D={D_{0^{e}}}^{-kt}$$
where $D_{0}$ is the initial degradation rate, *k* is the degradation constant, and *t* is time. This model helps align scaffold degradation with tissue healing, ensuring structural support during the critical initial phase of regeneration.

This approach allows for the seamless integration of the regenerated tissue with the patient’s existing structures, reducing the risk of complications and improving the overall quality of the repair [87].

## 4. Challenges

While the integration of biomimetic principles into rehabilitation technology offers significant promise, as presented above, several challenges must be addressed to fully realize its potential. These challenges range from the technical difficulties of translating biological systems into functional human-compatible technologies to ethical considerations surrounding the use of animal models in research [88]. Additionally, as biomimetic rehabilitation technologies advance, it is essential to ensure their accessibility and integration into clinical practice. This section discusses these challenges and outlines potential future directions for research and development in this interdisciplinary field.

### 4.1. Technical Challenges

One of the primary technical challenges in developing biomimetic rehabilitation technologies is the complexity of accurately replicating biological processes in mechanical or electronic systems. Biological systems, particularly those related to movement and sensory processing, are highly intricate, involving multiple interacting physiological, biochemical, and neural mechanisms [89]. Translating these systems into functional technologies for human rehabilitation requires a deep understanding of the underlying biological mechanisms and engineering expertise needed to recreate them effectively.

For instance, while robotic exoskeletons have made significant strides in mimicking the movement patterns of animals, achieving the same level of adaptability and energy efficiency remains a challenge. Animal locomotion, such as the spring-like movements seen in kangaroos or the efficient gait of quadrupeds, involves a level of fluidity and energy conservation that is difficult to replicate in mechanical systems [90]. Technologies must not only emulate these natural movements but also adapt to the unique anatomical and physiological characteristics of each human user. Developing actuators and sensors that can provide real-time feedback and adjust movements dynamically is critical for achieving this level of adaptation [91].

Additionally, predictive musculoskeletal simulations, while invaluable for understanding the principles of animal movement and applying them to rehabilitation, face limitations in accurately modeling the complexities of human anatomy across different populations. The scaling of musculoskeletal properties to match human physiology—especially when modeling variations in body size, posture, and gait speed—presents a significant challenge [12]. Ensuring that these models can predict movement accurately across a diverse range of individuals is necessary for their effective application in clinical rehabilitation.

### 4.2. Ethical Considerations

As biomimetic technologies move toward human applications, it is critical to address the ethical dimensions associated with their clinical use. Clinical trials and regulatory approvals should consider issues such as accessibility, equity, and the potential for health disparities. Expanding access to these advanced technologies across diverse patient demographics can mitigate inequalities and ensure that the benefits of biomimetic advancements are broadly distributed. This human-centered approach not only enhances the ethical foundation of biomimetic research but also improves the real-world applicability and societal acceptance of these technologies. The use of animal models in biomimetic research raises important ethical questions [88,92].

While animals provide invaluable insights into biological processes that can inform human rehabilitation technologies, their use in research must be carefully justified and conducted with the highest ethical standards. This includes ensuring that animal studies are designed to minimize suffering, using alternatives wherever possible, and adhering to the principles of the 3Rs—Replacement, Reduction, and Refinement [93].

#### 4.2.1. Replacement

Some studies have minimized animal use by employing in vitro models or computer simulations to replicate biological functions before advancing to in vivo animal studies [94]. For instance, predictive musculoskeletal simulations are used to model animal movement patterns and muscle dynamics, reducing the initial reliance on live animals by allowing the preliminary testing of robotic systems and prosthetics in a simulated environment.

Researchers often design experiments to obtain maximum data from a minimum number of animals. For example, studies on animal locomotion and gait analysis aim to use fewer animals by employing advanced imaging and sensor technologies, collecting more detailed biomechanical data per individual [95]. Additionally, techniques such as cross-over studies allow animals to serve as their own controls, further reducing the number needed.

#### 4.2.2. Reduction

By maximizing data collection from each subject, researchers can reduce the number of animals required for statistical validity. For instance, in studies examining gait patterns, sophisticated imaging and sensor technologies capture a comprehensive set of biomechanical data, thus reducing the need for larger sample sizes [96].

In some studies, animals serve as their own control group, allowing for comparative analysis within the same subject. This method not only reduces the number of animals needed but also minimizes variability, leading to clearer, more reliable data. For example, prosthetic limb testing in animal models often employs cross-over designs, where the same subjects experience different experimental conditions sequentially [97].

#### 4.2.3. Refinement

In studies involving sensor applications or gait analysis, non-invasive methods like external sensor placement reduce animal discomfort while allowing precise monitoring of movement [98]. Additionally, remote monitoring and non-contact imaging techniques (e.g., high-resolution video capture) limit physical handling, thereby minimizing stress and promoting well-being.

Certain animals, such as axolotls and zebrafish, are selected for their innate regenerative abilities, which can be studied with minimal intervention. This approach ensures that studies on tissue regeneration are less invasive, leveraging the species’ natural biology for scientific benefit without extensive harm [99]. Ethical protocols also mandate refined housing and environmental enrichment to reduce stress, promoting the overall welfare of these animals during their time in the study.

Moreover, as biomimetic technologies increasingly draw from species with unique regenerative or adaptive abilities, there is a risk of over-reliance on a small number of model organisms, which could have implications for biodiversity and conservation. For example, the axolotl, a critical model for regenerative studies, is also a species under threat in the wild [41]. Researchers must balance the need for scientific advancement with the responsibility to protect and preserve the species they study [100].

In addition to animal ethics, the application of biomimetic technologies in humans also raises questions about safety, efficacy, and accessibility. It is crucial to conduct thorough clinical trials to ensure that these technologies are safe and effective for all patients, and to address any potential disparities in access to advanced rehabilitation technologies. Ensuring that these technologies are available to a broad patient population, regardless of socioeconomic status, will be an important consideration as the field advances.

### 4.3. Translational Challenges

Another major challenge lies in the translation of findings from animal models to human applications [101]. Although animal studies provide crucial insights into the mechanisms of gait and neural regeneration, significant differences remain between animal and human biology that can limit the direct applicability of these findings. For example, the regenerative capabilities observed in animals like zebrafish or axolotls are not directly transferrable to humans due to fundamental differences in cellular and molecular processes [102]. This makes it difficult to develop therapies that can achieve similar levels of regeneration in human patients.

Similarly, the differences in gait between quadrupedal animals and bipedal humans can complicate the direct application of gait analysis findings. Studies that focus on mouse models of parkinsonism provide valuable information on how CNS dysfunctions affect gait, but translating these velocity-dependent changes into therapeutic interventions for human patients remains challenging.

Enhancing the accuracy of translational models is crucial in bridging these gaps. One approach involves improving cross-species biomechanical and neural simulation frameworks, which allow researchers to apply insights from animal models to human physiology with greater precision [103]. Predictive musculoskeletal simulations, for example, have emerged as valuable tools in understanding and adapting biomechanical differences across species. By accurately modeling variables like joint mechanics, muscle force, and gait patterns, these simulations make it possible to tailor assistive technologies, such as exoskeletons or prosthetics, to meet the specific physical demands of human users.

Further advancements can be achieved by developing hybrid models that integrate both animal and human data through AI-driven simulations. These hybrid models enable more accurate predictions of human responses to biomimetic devices, drawing on data from both human and animal studies to inform technology development [104]. Such models are particularly beneficial in personalizing rehabilitation strategies, as they allow for adjustments based on individual physiological data, thus enhancing the efficacy and comfort of wearable assistive devices.

Implementing adaptive technologies inspired by animal models offers another solution to cross-species translational challenges. Variable stiffness actuators in exoskeletons, for instance, replicate the natural adaptability seen in animal movement, allowing devices to adjust their rigidity based on the user’s specific needs and movements. Similarly, biohybrid materials for prosthetics—materials that mimic the flexibility and durability of natural tissues—help align the mechanical properties of these devices with human biomechanics, ensuring greater user comfort and natural functionality [105].

The integration of sensory feedback mechanisms, such as proprioceptive feedback in robotic systems, further enhances the adaptability of biomimetic devices. Proprioception, an essential sensory feedback system in animals, provides real-time information about body position and movement. By incorporating this feature into assistive devices, engineers can develop systems that react to the user’s actions with greater fluidity and responsiveness, effectively closing the loop between the user’s intentions and the device’s mechanical response. This integration allows devices to adapt to the user’s sensory environment, promoting natural interaction and supporting the motor learning processes essential for effective rehabilitation.

Bridging these gaps requires innovative approaches to adapt the core principles learned from animal models into human-compatible technologies, ensuring that the therapies and devices remain effective in a clinical setting [106]. Through a combined focus on cross-species modeling, adaptive technical solutions, and ethical scaling, biomimetic research can better bridge the gap between animal-based insights and effective, ethically implemented rehabilitation solutions for humans.

## 5. Future Directions

The future of rehabilitation technology lies in the continued fusion of biological insights with advanced engineering, leading to innovations that can better replicate and enhance natural movement and recovery processes. Addressing the challenges of integrating biomimetic principles into clinical practice will require a multifaceted approach, focusing on research, technological innovation, interdisciplinary collaboration, and the broader implementation of these advances in diverse healthcare settings. Here are several key areas for future development.

### 5.1. Advancing Computational Models, AI, and Simulations

The role of computational models and predictive simulations will continue to be central in bridging the gap between animal studies and human applications. As seen in studies that explore the relationship between body size, speed, and locomotion across species [12], advanced simulations can help researchers understand the fundamental principles that govern movement. Future work should aim to refine these models to better capture the complexities of human anatomy and physiology across a wide range of ages, body types, and health conditions.

One promising direction is the integration of machine learning and artificial intelligence (AI) with musculoskeletal modeling. Future research could focus on integrating AI to enhance the adaptability and personalization of rehabilitation programs. Key areas of exploration include the development of AI-driven models capable of the real-time adaptation to patient progress, creating personalized rehabilitation plans that adjust dynamically to the individual’s physical and cognitive states [107]. AI can analyze large datasets from both animal and human studies, identifying patterns and making predictions about movement that were previously difficult to quantify [108]. One other potential area involves integrating machine learning algorithms with wearable sensors inspired by animal proprioception systems. This AI-driven system could monitor and analyze real-time data on patient movements, adjusting therapeutic exercises based on immediate feedback. For instance, stroke patients could benefit from a system that detects minor gait asymmetries and automatically modifies the rehabilitation protocol to address specific weaknesses [109]. This approach could lead to more efficient, patient-centered rehabilitation with fewer in-person adjustments.

Another promising area for future research is the creation of predictive models that leverage AI to forecast patient recovery patterns. Drawing on animal studies of motor recovery, researchers could develop AI models trained on both animal and human data, allowing clinicians to anticipate individual responses to rehabilitation treatments. Such models could inform personalized therapy recommendations, aligning with each patient’s specific progress, reducing trial-and-error adjustments, and ultimately enhancing the effectiveness of rehabilitation interventions.

Finally, AI could support the development of hybrid bio-mechanical systems that combine biomimetic prosthetics or exoskeletons with adaptive feedback mechanisms. For example, incorporating AI-driven variable stiffness actuators could allow exoskeletons to adjust in real time to different terrains or physical tasks, mimicking the adaptability seen in animals. This integration would facilitate smoother, more natural movements, improving user comfort and effectiveness during daily activities.

### 5.2. Development of Hybrid Bio-Mechanical Devices

As the understanding of animal locomotion and neural plasticity deepens, there is potential to develop hybrid devices that combine biological and mechanical components. For example, the field of soft robotics, which takes inspiration from the flexible and adaptive movements of animals like octopuses and worms, offers new opportunities for creating exoskeletons and prosthetic devices that are both strong and highly adaptable [110]. These devices could be designed to provide varying levels of support, adjusting in real time to the user’s needs and the environment, whether they are navigating uneven terrain or moving at different speeds.

In addition to soft robotics, biohybrid systems that integrate living tissues with synthetic materials represent another area of growth [111]. These systems could mimic the responsiveness of biological muscles and tendons, providing a more natural interface between the human body and assistive devices. Research into biomimetic materials, such as those that replicate the energy-storing properties of animal tendons [112], could lead to prosthetics and orthotics that more closely emulate the dynamic behavior of natural limbs. This would not only improve user comfort but also reduce the energy required for movement, making daily activities more accessible for individuals with mobility impairments.

### 5.3. Expanding the Scope of Neural Regeneration Research

Building on insights from animals with remarkable regenerative capabilities, such as zebrafish and axolotls, future research should focus on developing therapies that enhance neural regeneration and plasticity in humans [113]. This could involve exploring the genetic and molecular pathways that enable these animals to regenerate spinal cord tissue and other nervous system components, with the goal of translating these mechanisms into clinical therapies.

Gene editing technologies like CRISPR-Cas9, which allow precise manipulation of genetic material, could be leveraged to activate regenerative pathways in human cells [114]. By identifying and replicating key genes involved in the regenerative processes of animals, scientists could potentially create new treatments for spinal cord injuries, strokes, and neurodegenerative diseases. These advances would complement existing rehabilitation technologies, offering a more holistic approach to restoring function after neurological damage.

### 5.4. Integrating Rehabilitation Technologies into Broader Healthcare Ecosystems

For biomimetic rehabilitation technologies to have a widespread impact, they must be integrated into broader healthcare ecosystems, including tele-rehabilitation and digital health platforms. The COVID-19 pandemic highlighted the importance of remote care, and this trend is likely to continue as healthcare providers seek ways to deliver high-quality care outside traditional clinical settings [115]. Future efforts should focus on creating digital platforms that support remote monitoring and adjustment of biomimetic rehabilitation devices, allowing patients to receive continuous feedback and support from their therapists.

These platforms could leverage data from wearable sensors, VR environments, and AI algorithms to provide personalized exercise programs, track patient progress, and offer real-time adjustments to rehabilitation protocols [4]. For instance, data on a patient’s gait patterns collected through a wearable device could be analyzed using AI to detect changes that may indicate improvement or the need for adjustments to their therapy plan [116]. This kind of integration would not only enhance the effectiveness of rehabilitation but also ensure that it remains accessible to those who live in remote or underserved areas.

### 5.5. Integration of Biomimetic Technologies with Digital Health Platforms

The combination of biomimetic technologies with digital health platforms represents an exciting avenue for advancing telerehabilitation and remote patient monitoring. By integrating wearable biomimetic devices, such as sensors inspired by proprioceptive systems or exoskeletons with adaptive feedback, with digital health platforms, clinicians can remotely monitor patient progress, adjust therapies, create digital twins models and track recovery in real time.

One potential application is in telerehabilitation for stroke patients or individuals with mobility impairments. Wearable sensors modeled after proprioceptive feedback systems could continuously monitor gait, balance, and muscle activity, transmitting these data to a digital platform accessible to healthcare providers [117]. This real-time monitoring would allow clinicians to assess patient progress, identify potential issues, and adjust rehabilitation protocols without requiring the patient to be physically present. Such integration would be especially valuable for patients in rural or underserved areas, who may have limited access to rehabilitation facilities.

Furthermore, remote monitoring of biomimetic devices, such as wearable exoskeletons or prosthetic limbs with AI-driven adaptability, could provide insights into patient adherence and device functionality in daily activities. For example, a patient’s activity level, movement patterns, and feedback from an exoskeleton could be monitored remotely, allowing adjustments to be made to the device’s settings or therapy plan as needed. This approach not only supports continuity of care but also enhances personalized rehabilitation by making adjustments based on real-world usage data [116].

The integration of these biomimetic technologies with digital health platforms could also support patient engagement. Gamified elements, progress tracking, and personalized feedback provided through mobile applications could motivate patients to adhere to their rehabilitation routines [115], further improving outcomes. Overall, the convergence of biomimetic technology and digital health holds great promise for making rehabilitation more accessible, adaptable, and personalized.

### 5.6. Strengthening Interdisciplinary Collaboration

The development of biomimetic rehabilitation technologies requires collaboration across multiple fields, including biomechanics, neuroscience, materials science, robotics, and clinical rehabilitation. By fostering stronger interdisciplinary partnerships, researchers can accelerate the pace of innovation and ensure that new technologies are designed with both scientific rigor and practical usability in mind [118]. Collaborative efforts can also bridge the gap between fundamental research on animal models and the development of user-friendly, patient-centered technologies.

This collaboration should extend beyond academia to include partnerships with industry and healthcare providers. By involving end-users—patients and clinicians—in the design process through methods like co-creation and design thinking, developers can create technologies that are better suited to real-world needs. Engaging with healthcare professionals during the early stages of development can also ensure that new technologies align with clinical workflows, making them easier to integrate into existing care practices.

## 6. Conclusions

The challenges faced in the development of biomimetic rehabilitation technologies are significant, but so too are the opportunities. By continuing to draw inspiration from the natural world, researchers can develop new tools and therapies that enhance the recovery process for patients with neurological and musculoskeletal impairments. Overcoming the technical, ethical, and practical challenges will require a concerted effort from multiple disciplines, but the potential benefits for patient care are immense. As the field of biomimetics continues to evolve, its application in rehabilitation science holds the promise of transforming the future of healthcare.

## Figures and Tables

**Figure 1 biomimetics-09-00723-f001:**
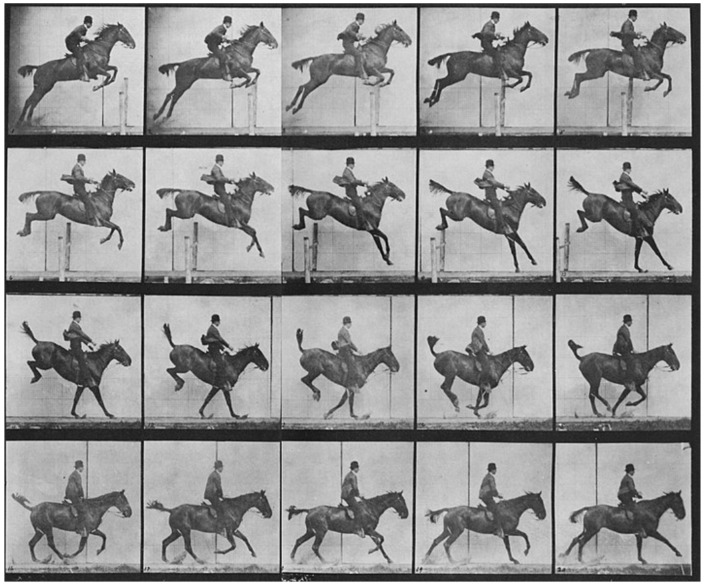
Horse running by Muybridge Eadweard. Credit: Wellcome Collection gallery (https://openartimages.com/search/eadweard-muybridge (accessed on 22 March 2018)), CC-BY-4.0.

**Figure 2 biomimetics-09-00723-f002:**
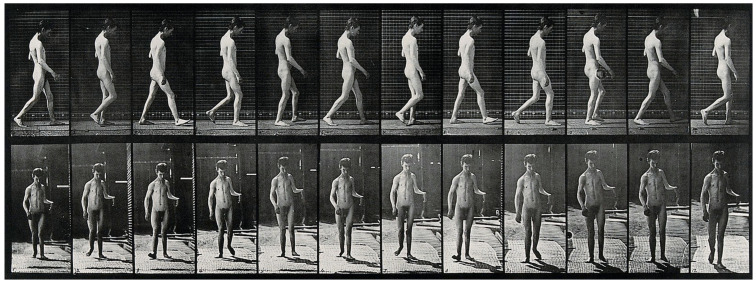
A man walking. Photogravure after Eadweard Muybridge, 1887. Credit: Wellcome Collection gallery (https://openartimages.com/search/eadweard-muybridge (accessed on 22 March 2018)), CC-BY-4.0.

**Figure 3 biomimetics-09-00723-f003:**
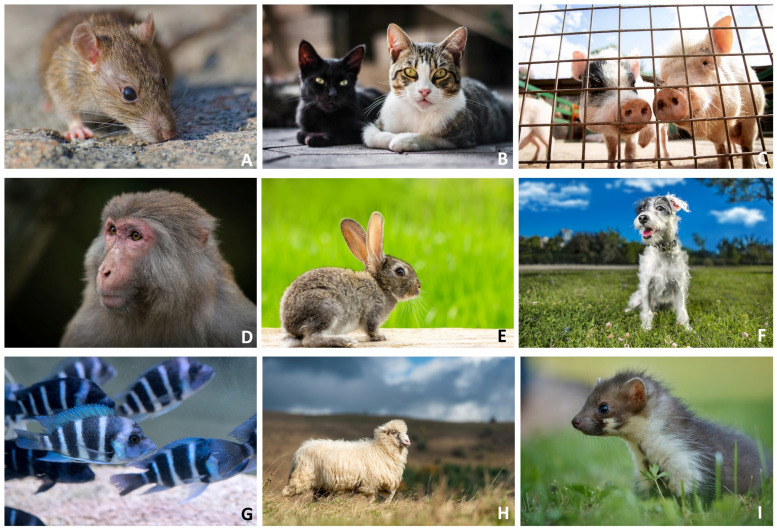
Example of animals inspiring research in technology-supported rehabilitation. (**A**) Mice for motor recovery; (**B**) cats for robotic assisted gait training; (**C**) pigs for post-injury muscular recovery; (**D**) macaques for cognitive training; (**E**) rabbits for tendinitis and tendon recovery; (**F**) dogs for physical therapy techniques; (**G**) zebrafish for nerve regeneration; (**H**) sheep for osteoarthritis; (**I**) rats for nerve recovery and sensorimotor function (Photos from Freepik https://www.freepik.com/ (accessed on 10 November 2024)).

**Figure 4 biomimetics-09-00723-f004:**
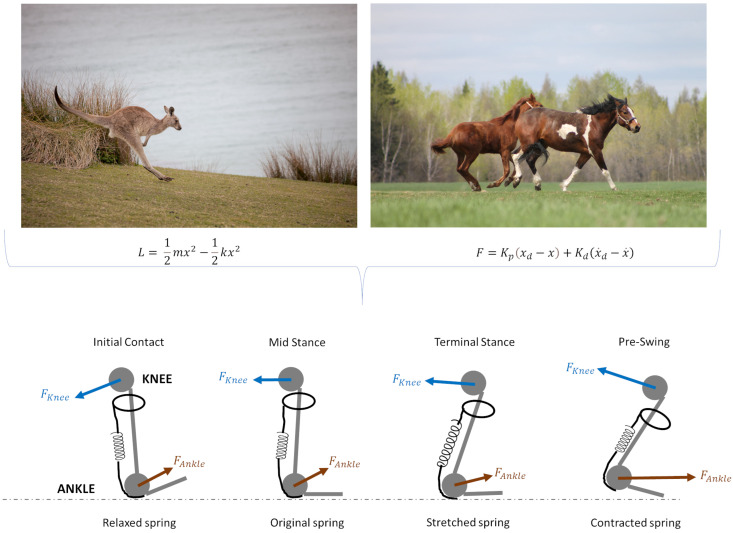
Comparative illustration of elastic and mechanical force models in animal locomotion applied to soft exoskeleton. Top figures show examples of animal movement dynamics: a kangaroo (**left**) illustrating the role of elastic energy storage and release during hopping, and horses (**right**) showcasing controlled movement dynamics with muscle forces. Bottom figures represent the simplified biomechanical model of joint forces and elastic components at different stages of the gait cycle, specifically highlighting the knee and ankle forces. During initial contact, the spring (representing elastic tissue) is relaxed as ground reaction forces are absorbed. In mid stance, the spring maintains its original state, supporting joint stability. Terminal stance shows a stretched spring, maximizing energy storage in preparation for the push-off phase. In pre-swing, the spring contracts, converting stored energy into propulsion. Arrows indicate the directions of knee and ankle forces applied during each phase, depicting the interaction between muscle forces and elastic tissue response (Photos from Freepik https://www.freepik.com/ (accessed on 10 November 2024)).

**Table 1 biomimetics-09-00723-t001:** Examples of animal studies in general healthcare research.

Animal Model	Research Area	Purpose/Insight	Examples/Applications
Mice	Immuno-Oncology [19]	Study tumor growth, metastasis, and response to treatments due to their similar genetic and physiological traits to humans.	Development of immunotherapies and targeted treatments for cancers like breast cancer.
Rats	Neurology [20]	Used to model neurodegenerative diseases, helping researchers understand brain function and disease progression.	Testing new drugs for Parkinson’s and exploring potential treatments for brain injuries.
Guinea Pigs	Respiratory Disorders [21]	Models for studying asthma, chronic obstructive pulmonary disease (COPD), and tuberculosis due to similar respiratory systems.	Development of inhalers, vaccines, and medications for asthma and respiratory conditions.
Rabbits	Cardiovascular Medicine [22]	Used for studying atherosclerosis and heart disease because of their susceptibility to cholesterol-induced heart conditions.	Development of heart disease treatments and cholesterol-lowering drugs.
Non-Human Primates (e.g., macaques)	Infectious Disease Research [23]	Study immune responses and potential vaccines for viruses like HIV/AIDS and COVID-19, closely mimicking human immune response.	Development of HIV vaccines, testing efficacy of new antiviral drugs.
Zebrafish	Developmental Biology [24]	Study of vertebrate development, genetic functions, and effects of genetic mutations, due to transparent embryos and fast reproduction.	Insights into genetic disorders, drug screenings, and developmental processes.
Pigs	Organ Transplantation [25]	Used in transplantation studies due to anatomical and physiological similarities to humans, such as heart and liver structure.	Research into xenotransplantation and testing new surgical techniques for organ transplants.
Sheep	Orthopedics and Regenerative Medicine [26]	Studying bone healing, joint replacement, and tissue regeneration, reflecting similar bone size and healing processes as humans.	Development of joint replacement materials and testing new surgical procedures.
Dogs	Diabetes Research [27]	Used for studying type 1 diabetes due to their similar insulin response mechanisms.	Development of insulin therapies and devices for glucose monitoring.
Ferrets	Influenza and Respiratory Viruses [28]	Used to model respiratory infections as they show similar symptoms to humans when infected.	Study of flu virus transmission and vaccine development for influenza.

## Data Availability

No new data were created or analyzed in this study. Data sharing is not applicable to this article.

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
