# Peer review of "Animals as Architects: Building the Future of Technology-Supported Rehabilitation with Biomimetic Principles"

_biomimetics, 2024, doi:10.3390/biomimetics9120723_

Round 1
Reviewer 1 Report
Comments and Suggestions for Authors
The article examines the innovative use of biomimetic principles in rehabilitation technologies. It highlights the role of animal models in inspiring the design of assistive devices such as robotic exoskeletons and prosthetics, which aim to mimic natural processes to enhance recovery. Despite its comprehensive approach, the article could benefit from several modifications to improve its impact and clarity. Here are some specific suggestions:1. The article would benefit from the inclusion of visual aids, such as diagrams and charts, to help illustrate complex concepts like the mechanics of animal-inspired technologies, making the content more accessible to readers.2. While the role of animal models is discussed, the article could elaborate more on the specific criteria for their selection and the unique contributions they make to the design of rehabilitation technologies.3. The section on technical challenges could be expanded with specific examples of engineering solutions that have been developed to overcome difficulties in replicating animal movement patterns in robotic systems.4. The discussion on ethical considerations would be more impactful if it included concrete examples of how ethical challenges have been addressed in previous studies, such as the application of the 3Rs (Replacement, Reduction, Refinement) in animal research.5. The article could provide more detailed case studies or examples of how biomimetic devices have been successfully implemented in clinical settings, demonstrating their impact on patient outcomes.6. Future research directions could be outlined more clearly, with specific goals or projects highlighted, particularly in emerging areas like AI integration and personalized rehabilitation programs.7. The integration of biomimetic technologies with digital health platforms could be further explored, discussing their potential role in tele-rehabilitation and remote patient monitoring.
Author Response
See attached document

Reviewer 2 Report
Comments and Suggestions for Authors
This paper explores the application of biomimetic principles in rehabilitation technologies, highlighting how animal models can inspire the design of assistive devices and improve rehabilitation outcomes. The topic is innovative and of significant relevance in the interdisciplinary field of technology and medicine. By integrating biomimetic approaches, the paper demonstrates the potential for substantial advancements in rehabilitation sciences.
The paper lacks visual materials, such as figures or diagrams, which could help readers better understand the complex technical concepts and biological mechanisms simulated and applied in rehabilitation technologies. It is recommended to include relevant visuals, such as explanations of biomimetic principles, design schematics of devices, or photographs of animal models.
The discussion of certain technical details is rather brief. For instance, while energy-efficient exoskeletons, prosthetics, and wearable sensors are mentioned, there is a lack of detailed descriptions of their specific implementation technologies and biological prototypes.
Although the paper mentions the challenges of applying animal research findings to human applications, there is not enough in-depth discussion on methods to overcome these challenges, particularly concerning ethical and technical implementation aspects.
Including specific case studies or examples of successful applications would make the paper more convincing and practical.
Author Response
See attached document

Round 2
Reviewer 1 Report
Comments and Suggestions for Authors
The author have satisfactorily addressed all the concerns and suggestions raised in the previous round of review. The revised manuscript has been significantly improved in terms of clarity, organization, and depth of analysis. The figures and tables are now of publishable quality. The writing and formatting also meet the journal's standards after the revision. I believe the current version of the manuscript is suitable for publication in the journal
Author Response
Dear Reviewer,
Thank you again for taking the time to evaluate this revised version and for approving it for publication.
Best regards,